# Recent Advances on Development of Hydroxyapatite Coating on Biodegradable Magnesium Alloys: A Review

**DOI:** 10.3390/ma14195550

**Published:** 2021-09-24

**Authors:** Junxiu Chen, Yang Yang, Iniobong P. Etim, Lili Tan, Ke Yang, R. D. K. Misra, Jianhua Wang, Xuping Su

**Affiliations:** 1Key Laboratory of Materials Surface Science and Technology of Jiangsu Province, Changzhou University, Changzhou 213164, China; xingyan622@163.com (Y.Y.); wangjh@cczu.edu.cn (J.W.); sxping@cczu.edu.cn (X.S.); 2Jiangsu Collaborative Innovation Center of Photovoltaic Science and Engineering, Changzhou University, Changzhou 213164, China; 3Institute of Metal Research, Chinese Academy of Sciences, Shenyang 110016, China; ini2etim@yahoo.com (I.P.E.); lltan@imr.ac.cn (L.T.); 4Department of Metallurgical, Materials and Biomedical Engineering, University of Texas at El Paso, 500 West University Avenue, El Paso, TX 79968, USA; dmisra2@utep.edu

**Keywords:** magnesium alloy, hydroxyapatite, coating, degradation

## Abstract

The wide application of magnesium alloys as biodegradable implant materials is limited because of their fast degradation rate. Hydroxyapatite (HA) coating can reduce the degradation rate of Mg alloys and improve the biological activity of Mg alloys, and has the ability of bone induction and bone conduction. The preparation of HA coating on the surface of degradable Mg alloys can improve the existing problems, to a certain extent. This paper reviewed different preparation methods of HA coatings on biodegradable Mg alloys, and their effects on magnesium alloys’ degradation, biocompatibility, and osteogenic properties. However, no coating prepared can meet the above requirements. There was a lack of systematic research on the degradation of coating samples in vivo, and the osteogenic performance. Therefore, future research can focus on combining existing coating preparation technology and complementary advantages to develop new coating preparation techniques, to obtain more balanced coatings. Second, further study on the metabolic mechanism of HA-coated Mg alloys in vivo can help to predict its degradation behavior, and finally achieve controllable degradation, and further promote the study of the osteogenic effect of HA-coated Mg alloys in vivo.

## 1. Introduction 

Magnesium (Mg) alloys, as the third generation of medical materials, have good biocompatibility, bioactivity, and biodegradability, which have become a research hotspot in the field of biomaterials in recent years. Mg alloys have the characteristics of being close to the density and elastic modulus of human bone, which can effectively avoid the “stress shielding” effect caused by the elastic modulus mismatch [1,2], so it has a good application prospect in the field of orthopedics [3]. However, the degradation rate of Mg alloys is fast, which generates a high alkaline environment, leading to the occurrence of bone lysis. In addition, the bubbles generated in the degradation process are attached to the surface of the implant, which usually hinders the adhesion and growth of cells on the implant surface, and delays the healing of the surgical site [2]. As a bone filling material, the osteoinduction and osteogenic properties of Mg alloys need to be further improved [4].

Coating with bioactive materials is an effective method to improve the biological properties of implants. Hydroxyapatite (HA) is an ideal implant coating material because its chemical composition and properties are similar to the main inorganic components of human bone and teeth. HA coating on the surface of Mg alloys has the following advantages: reducing the degradation rate of Mg alloys and improving the biological activity of Mg alloys [5]; having excellent bone conductivity and biological activity, and then quickly fusing with host bone tissue [6]; and biodegradable, with the ability of bone induction and bone conduction, which is beneficial for orthopedic applications [7]. However, the shortcomings of HA coating, such as high brittleness and low strength, make it difficult to be applied to the load-bearing parts. In order to improve these shortcomings, HA biological composite coating is usually used [8,9,10,11]. HA coating on the surface of Mg alloys is evaluated in the following four ways: First, the in vitro degradation of the coating samples is determined by electrochemical experiments and in vitro immersion, which provides a reference for the practical application of the coating. Second, the bonding strength between the coating and the sample is measured. Bonding strength has a significant role in the service life of the coating, which is the premise of the protective property of the coating. The binding strength mainly depends on the amount of interfacial binding force, which is the force required to strip the coating, per unit area, from the substrate [12,13]. The initial adhesion and accumulation of bacteria usually leads to biofilm and the failure of biomaterial implantation. Therefore, the prevention of early bacterial infection after implantation is essential. The high antibacterial rate means that the sample does not cause an inflammatory response after entering the body [14]. In the cytotoxicity test, the higher the relative proliferation rate of the cells, the higher the cell viability. Finally, the activity of MC3T3-E1 cells on HA-coated Mg alloys is determined by the MTT method, to determine the osteogenic properties of the alloy. Sometimes, the biomaterials and their surrounding tissues coexist and produce synergistic effects. The process mainly depends on the compatibility of the artificial biomaterials, such as osteogenic capability and antibacterial capability. Surface modification can effectively improve compatibility, by changing the surface chemistry, microstructure, and other materials attributes [14].

In this review, recent advances in the development of HA coatings on biodegradable Mg alloys are summarized. Different HA coating preparation methods, and the effects of HA coatings on the mechanical properties, corrosion resistance, biological activity, and biocompatibility of Mg alloy are reviewed in detail. This review will provide a guideline to design HA coating on Mg alloys with better bioactivity.

## 2. Preparation and Properties of HA Coating

At present, the commonly used preparation methods mainly include spraying, electrochemical process, micro-arc oxidation, sol-gel, hydrothermal synthesis, etc. The following will introduce these preparation methods of HA coating on surfaces of Mg alloys, and their effects on the properties of Mg alloys in detail.

### 2.1. Spraying

Spraying is a kind of surface strengthening technology, an important part of surface engineering technologies. The preparation of HA coating on the surface of Mg alloys is mainly composed of thermal spraying and cold spraying. The temperature of thermal spraying is relatively high, and the main methods include explosion spraying, arc spraying, plasma spraying, and supersonic flame spraying. The temperature of cold spraying is lower, and the spraying speed is fast, which can prepare a dense, thick coating.

#### 2.1.1. Thermal Spraying

Thermal spraying is a method of coating fabrication, which uses flame, arc, plasma, compressed gas, etc., to heat and accelerate linear or powdery material, and then form molten, semi-molten, or unmelted solid, high-speed particles flowing to the substrate, with constant deposition. Among all the thermal spraying methods, plasma spraying is the most commonly used method to prepare HA coating.

Gao et al. [15] prepared HA coating on the surface of an AZ91HP Mg alloy, by plasma spraying. The average corrosion rate of the alloy was 0.11 mm/y in the simulated body fluid (SBF) solution, while that of the HA-coated alloy was reduced down to 0.017 mm/y. The contact angle of the coated alloy (30 ± 3°) was less than that of the base alloy (46 ± 1°). A contact angle less than 60° indicates a hydrophilic surface with better cell adhesion [16,17]. The surface energy of the coating is higher than that of the Mg alloy substrate, and the adhesion of cells on the coating is improved, which is beneficial to the growth of cells. At the same time, a prothrombin time (PT)-measuring experiment was performed on the alloy before and after coating. With increasing PT, the clotting factor in the coagulation system decreases, and the anticoagulant activity of the materials is better. The test result shows that the PT of the coating was 18 s, but the PT of the Mg alloy was 11 s, which indicates that HA coating could significantly improve the anticoagulant activity of Mg alloys. The HA composite coating prepared by thermal spraying could further improve the corrosion resistance of the coating, where the Mg alloy was plasma-sprayed with pure HA and HA reinforced with zinc oxide (ZnO) powder at three levels (4, 8, and 12 wt.%). With an increase in the ZnO content, the surface microhardness of the Mg alloy increased significantly. The microhardness of the HA + 12% ZnO coating reached 280 HV. The previous studies also proved that the surface hardness of the composite HA coating was higher than that of the pure HA coating [18,19]. The coated alloy showed better hydrophilicity and corrosion resistance [20]. In a further study, the Mg alloy was coated with an HA–Nb composite. The Nb-reinforced HA (HA–Nb) coating was obtained through plasma spraying, by varying the weight percent (wt.%) of Nb in HA at three levels (10, 20, and 30 wt.%). Compared with pure HA coating, HA–Nb coating had a stronger inhibition effect on the corrosion of the Mg alloy. With a progressive increment in the Nb content in HA, the protection efficiency (Pe) for HA–Nb coatings was increased (~4, 12, and 18%), in comparison to the pure HA coating. The HA–Nb coatings exhibited no adverse effect on the erythrocytes, and the hemolysis rate (HR) was within the domain of safe values (<5%) for implant materials [21]. However, there are still problems for plasma spraying, for example, due to the high temperature of spraying, the HA coating would produce an amorphous phase after spraying, which might affect the interface integrity of the bone implant. Strong reabsorption of the amorphous phase may cause mechanical and bonding instability of the coating, which limits its application [22].

In order to overcome the thermal decomposition of HA, caused by the high temperature during spraying, Mardali et al. [23] employed a high-speed oxygen fuel (HVOF) spraying method, with good thermal stability, to study the deposition of HA on Mg alloys, in comparison with the traditional flame-spraying method. The results showed that the HVOF coating produced a more stable phase and less surface roughness, but more cracks. The electrochemical impedance spectroscopy (EIS) test showed that the corrosion resistance of HVOF coating was higher than that of flame-spraying coating. Over time, there are many cracks in the samples, which led to corrosion aggravation. However, the corrosion products formed on the traditional flame-sprayed samples in SBF became a protective layer, which increased the corrosion resistance in SBF. Compared with plasma spraying, HVOF could avoid the decomposition of HA, to some extent, but it had a negative effect on the long-term corrosion resistance of the Mg alloy matrix.

#### 2.1.2. Cold Spraying

Cold spraying is a new surface technology, developed based on the aerodynamics principle of the alloy. A gas (air, nitrogen, helium, and argon as the accelerating medium) is heated and pressurized to produce supersonic airflow (300–1200 m/s), which drives the spraying powder to accelerate to a supersonic state in the solid state, thus dragging metal particles to impact the substrate, causing strong plastic deformation, and depositing on the surface of the substrate to form a coating [24]. As an emerging surface technology, high-pressure cold-spraying technology relies on high speed rather than high temperature, which can effectively avoid problems such as coating oxidation, phase transformation, and poor crystallization that are caused by high temperature. It has many potential applications and can bring greater economic benefits to the industry. Yao et al. [25] prepared a novel HA/Zn composite coating on the surface of an AZ91 Mg alloy, by cold spraying. The HA/Zn ratios in weight were 3:7 (Z7H3) and 7:3 (Z3H7), respectively. As shown in Figure 1, the coating has a compact structure and is well combined with the substrate. Meanwhile, the corrosion current and corrosion rate of the coated Mg alloy were decreased by about four times. Cold-sprayed HA coatings can improve the corrosion resistance of the Mg alloy matrix. However, there are few studies on the degradation performance of the cold-sprayed HA coating and its regulation mechanism on osteoblasts.

### 2.2. Electrochemical Process

#### 2.2.1. Electrodeposition

Electrodeposition is a process by which a metal or alloy is deposited from the aqueous, non-aqueous, or molten salt of its compound. The process of producing HA coating by electrodeposition is simple and does not need complicated equipment. In addition, electrodeposition can be applied to porous or heterogeneous substrates, resulting in high crystallinity and low residual stress of HA coatings [26].

Uddin et al. [27] prepared an HA coating on an AZ31B Mg alloy surface, by electrodeposition, and investigated the effect of current density on the formation, properties, and degradation of the coating. The results showed that the HA coating deposited at a low current density was denser and more uniform than that deposited at a high current density. It had a higher corrosion potential, higher current, higher polarization resistance, less damage in the corrosion area, and thus better corrosion resistance. In order to further improve the coating performance, Uddin et al. [28] performed deep ball polishing on the AZ31B Mg alloy samples before electrodeposition of the HA coating. The surface of the polished Mg alloy could deposit about 110 μm of dense sheet HA crystal, which was thicker than the surface of the untreated Mg alloy (about 90 μm). The adhesion strength of the polished HA coating (13.2 MPa) was also higher than that of the untreated HA coating. Compared with the untreated sample, the corrosion current density of the polished HA coating samples in SBF solution decreased by six times and the corrosion potential increased by 1.44 times, which significantly improved the corrosion resistance. A cell viability test of human fibroblast cells (HFFF2, Sigma Aldrich) was conducted, to investigate the cytotoxicity of the treated surfaces. In order to evaluate the cell growth and proliferation, cell adhesions on the untreated and treated Mg alloy surfaces were observed under fluorescence microscopy, using nucleus staining (DAPI), after day 2. As shown in Figure 2a, the HA coating significantly promoted cell activity and growth compared with the uncoated samples. Figure 2b depicts the fluorescence images showing the presence of the nucleus of the cells (blue color) on the surface after day 2. The untreated and burnished surface exhibited a relatively sparse and comparable cell density. However, the two HA-coated samples showed a high concentration of cells spread over the entire surface, indicating enhanced fibroblast cell proliferation on the HA surface. Macrophages are produced by the differentiation of monocytes, in response to an infection. They secrete an array of cytokines that aid the host defense, tissue repair, and immunoregulation. An enzyme-linked immunosorbent assay (ELISA) was conducted to quantify the pro-inflammatory (TNF-α) and anti-inflammatory (arginase) cytokines produced by the macrophages that were cultured on the untreated and treated Mg alloy samples. The HA-coated samples favored expressing a significantly lower number of TNF-α and a high number of arginases, which proves that the Mg alloy that was treated by burnishing and HA coating would not pose any threat to the potential inflammation once it was inserted into the human body. Studies have shown that implants with enough rough surface could improve surface protein absorption, and thus enhance tissue/bone adhesion to the implants [29,30]. HA nano-coating was electrodeposited on the surface of the mechanical attrition-treated (SMATed) AZ31 Mg alloy. The roughness of the SMATed surface has greater surface activity, which is conducive to the deposition of molecules, greatly increasing the thickness of the coating, effectively preventing the penetration of corrosive media, and improving the bonding strength of the coating. The SMATed surface could effectively induce the nucleation of HA, making SMATed + HA samples have better biological activity [31].

Compared with the above mechanical treatment of the substrate surface, Rahman et al. [32] anodized a WE43 alloy in sodium hydroxide solution, and then coated the alloy with dicalcium phosphate dihydrate (CaHPO_4_·2H_2_O, DCPD) and HA successively. The results showed that after alkali treatment, the DCPD coating was transformed into a dense and crack-free HA coating, with a platy structure. Because the inorganic compositions of natural bone are similar to HA crystals, the plate-like structure of HA was conducive to the growth of bone tissue [33]. As can be observed from Table 1, compared with other samples, the microhardness and binding strength of the samples coated with HA were significantly improved. Not only can HA improve the mechanical properties of the substrate, but it can also improve the stability of the coating. In all the samples, the HA-coated WE43 alloy had a low hydrogen evolution rate, low degradation rate, and excellent corrosion resistance in the modified simulated body fluid (M-SBF). In addition to alkali treatment, HA coating was deposited on the fluoride-coated Mg-2.1Zn-0.22Ca alloy, by the pulsed reverse current electrodeposition method. Meanwhile, the microstructure and corrosion behavior of the composite coating were studied in comparison with those of conventional electrodeposition. Compared with the traditional galvanostatic electrodeposition coating, this composite coating presented a dense and uniform nano-rod structure. The corrosion current density of the coating decreased from 5.72 × 10^−5^ A/cm^2^ of the original alloy to 4.32 × 10^−7^ A/cm^2^, indicating that the corrosion resistance was increased by nearly two orders of magnitude [34].

Dehghanian et al. [35] prepared HA coatings, with different Si contents, on a Mg-5Zn-0.3Ca alloy, by the reverse pulsed current electrodeposition process. The substitution of SiO_4_^−4^ instead of PO_4_^3−^ ions in the HA structure caused an increase in pH for the substrate, during the electrodeposition process, which provided an appropriate state for coating nucleation and an increase in density for the Si-HA coatings. The Si-HA coating with a higher density could act as a resistant barrier to prevent SBF from penetrating into the substrate, thus improving the corrosion resistance of the alloy. Methyl thiazolyl tetrazolium (MTT) colorimetry was used to detect cell viability, and the direct contact method was used to detect cell adhesion. The results showed that the higher corrosion resistance of Si-HA coatings made their extracts more suitable for cell proliferation, and the MG63 cells had better proliferation in the HS2 coating (the mole ratio of SiO_3_^2−^/(PO_4_^−3^ + SiO_3_^2−^) was 0.2) extract, in comparison to that in the HA coating extract. Cells were attached to the surfaces of all the samples, and some cells spread across the surface and were in contact with each other. In order to further determine the optimal pulse process parameters, to obtain better performance of the coating, the Si-HA coating was deposited on the Mg-5Zn-0.3Ca alloy by the pulse electrodeposition method. The results showed that the Si-HA coating had good uniformity, adhesion, and density, with parameters of duty cycle of 0.1, current density of 40 mA/cm^−1^, pH of 5, and temperature of 85 °C [36]. The above studies indicate that the properties of HA coatings can be improved by changing the electrodeposition parameters, physical or chemical treatment of the Mg alloy substrate before electrodeposition, and changing the composition of the coatings, etc.

#### 2.2.2. Electrophoretic Deposition

Electrophoretic deposition is a process in which colloidal particles are deposited onto materials in a stable suspension, using a direct current electric field. Tayyaba et al. [37] used calcium hydroxide powder as the calcium source and orthophosphoric acid as the phosphorus source to prepare HA by the co-precipitation method, then electrophoresis deposition was performed on an extruded ZK60 alloy. The degradation rate of the ZK60 alloy in Ringer’s solution was decreased by five times, owing to an HA coating with a thickness of 15 μm. The HA and bioglass composite coating (HA-BG) has attracted many researchers’ interest, because of its significant substitution effect in the fields of maxillofacial surgery, orthopedics, tissue engineering, and so on. Chitosan (CS) has been widely used in biotechnology, because of its high biocompatibility, chemical stability, mechanical strength, and antibacterial properties. Singh et al. [38] deposited HA-BG-Fe_3_O_4_-CS composite coating with different concentrations of Fe_3_O_4_ nanoparticles (1, 3, and 5 wt.%) on an AZ91 Mg alloy. The results showed that the HA-BG-Fe_3_O_4_-CS coating had good adhesion to the substrate and there was no crack on the alloy surface. Due to the presence of Fe particles, there was a large number of spherical aggregates, composed of fine particles, on the coating surface. The composite coating had a porous structure, which improved the wettability of the coating, thus promoting cell adhesion and proliferation. Figure 3a shows the hemolysis rate (HR) of the different samples. The hemolysis rate of the coated samples was less than 5%, meeting the requirement on implant material. The morphology of the red blood cells on the sample is shown in Figure 3b. A large number of hemolytic red blood cells were found on the uncoated samples. The HA-BG coating could reduce the number of hemolytic red blood cells, and almost all the red blood cells interacting with the Fe_3_O_4_-doped coating maintained their original shape (approximately round). The coating could effectively inhibit the hemolysis of red blood cells after implantation and showed good compatibility with human blood. The crystallinity of the HA composite coating was better than that of the single HA coating. The current density of the 1 wt.% Fe_3_O_4_-coated sample was decreased by an order of magnitude [39]. Arangari et al. [40] prepared an HA coating containing carboxymethyl cellulose (CMC) and graphene (Gr) on an AZ31 Mg alloy, by two-step cathode electrophoresis deposition. They used an AZ31 alloy sample as the cathode and graphite as the anode (with similar surface areas) for deposition. The first deposition was carried out at a constant voltage of 50 V for 3 min, and then dried at room temperature for 20 min. The second deposition was carried out at a voltage of 30 V and the deposition time was extended to 5 min. It was found that the HA-CMC-Gr coating possessed the highest impedance and the lowest corrosion current density. The Gr in the coating prevented the corrosion solution from entering the AZ31 Mg matrix. CMC not only improved the bonding strength between the substrate and coating, but also improved the bonding strength among the HA particles, which made the structure of the HA coating denser. This denser and more complete coating significantly inhibited the diffusion of solution ions in the coating, thus reducing the corrosion rate and improving the corrosion resistance of the coating. The binding strength is an important factor to determine the mechanical properties of the coating, which reflects the expected life of the coating in medical applications. By comparing the HA coating with the HA-CMC coating, it was found that the addition of CMC increased the coating binding force from 1.06 MPa to 1.62 MPa. The SEM image of the coating along the cross-section of the substrate also showed a uniform coating without peeling, indicating that the coating quality was good. The presence of Gr increased the elastic modulus of the coating by about 33%, and the coating possessed good mechanical properties. The electrophoretically deposited HA composite coating showed good surface properties and corrosion resistance, which makes it promising for use in Mg-based implants.

### 2.3. Micro-Arc Oxidation

Micro-arc oxidation (MAO) is also known as plasma oxidation or anode spark deposition. MAO technology is mainly a combination of electrolyte and corresponding electrical parameters, and the instantaneous high temperature and high pressure generated by arc discharge on the surface of the Mg alloys, to form a ceramic film based on the metal oxide of the matrix, in order to improve the corrosion resistance and surface hardness of the Mg alloys [41].

Chaharmahali et al. [42] studied the effects of nanometer HA coatings, prepared by MAO with different HA contents (5, 10, 15 g/L), on the corrosion resistance of an AZ31B Mg alloy. Compared with the uncoated alloy, the polarization resistance of the coated alloy was averagely increased by 62%. The HA coating with a content of 15 g/L had the highest positive corrosion potential (−1.54 V) and the lowest corrosion current density (1.99 × 10^−6^ A/cm^2^). The HA coating prepared by MAO could effectively improve the corrosion resistance of the substrate. However, in the practical application, the composite HA coating had a better protective effect on the substrate than the single HA coating. Meanwhile, other elements in the composite coating could also make the coating have better mechanical properties and biological activity. Biodegradable HA/graphene oxide (GO) composite coatings were prepared on the surface of Mg alloys, by single-step MAO. The SEM observations in Figure 4 show that the presence of HA/GO could seal part of the pores on the coating and reduce the size of the parts of the pores. The contents of C and Ca in the HA/GO coating were increased significantly with even distributions, and the HA/GO was integrated into the surface structure during the oxidation process. The average bonding strength (40.7 MPa) between the coating and the Mg alloy matrix was also higher than that of the natural cortical bone (~35 MPa). Electrochemical tests showed that the corrosion current of the HA/GO-coated samples was two orders of magnitude lower than that of the uncoated Mg alloys. The polarization resistance of both the HA/GO coating and MAO coating was higher than that of the uncoated Mg alloy, which could effectively prevent the entry of corrosive ions. Especially, the polarization resistance of the HA/GO coating was higher than that of the MAO coating, indicating that the HA/GO coating had higher corrosion resistance [43]. A thick coating could better protect the substrate from corrosion by the biological environment [44]. Xiong et al. [45] prepared Ti_3_O_5_-HA ceramic coatings in the silicate electrolyte, with different concentrations of potassium fluorotanate (K_2_TiF_6_) and nanometer HA, using MAO technology. The thickness of the coating was increased by 2–11 μm compared with that of the pure HA coating, and the coating gradually became thicker with the increase in K_2_TiF_6_ concentration in a certain time range. The coating with a K_2_TiF_6_ concentration of 10 g/L showed the best corrosion resistance. The results showed that the surface of the coating was not corroded after soaking for 140 h, and the thickness of the coating was increased by 2 μm after soaking for 280 h. The polydopamine (PDA) coatings could provide corrosion protection on the surface of Mg alloys [46]. Feng et al. [47] used MAO-HA as the inner coating and PDA film as the outer coating on the surface of the Mg-Zn-Ca alloy. The PDA coating uniformly overlayed the MAO-HA coating, with good wettability. Compared with the single MAO-HA coating, the corrosion potential of the PDA/MAO-HA-coated sample was increased by 150 mV, and the corrosion current was decreased from 2.09 × 10^−5^ A/cm^2^ to 1.46 × 10^−6^ A/cm^2^. It was further confirmed that the PDA/MAO-HA-coated sample had a lower hydrogen evolution rate (4.40 mm/y), indicating that the PDA/MAO-HA coating effectively improved the corrosion resistance of the Mg-Zn-Ca alloy. The MTS (3-(4,5-Dimethylthiazol-2-yl)-5-(3-carboxymethoxyphenyl)-2-(4-sulfophenyl)-2H-tetrazolium) assay test was performed on the samples. As can be observed from Figure 5a, the absorbance of the PDA/MAO-HA-coated samples was the highest, which was significantly higher than that of the tissue culture plate (TCP) group. With an extension of the culture time, MC3T3-E1 cells proliferated rapidly in the PDA/MAO-HA extract, which resulted in malnutrition of the extract and reduced the cell activity. The results still showed that the PDA/MAO-HA coating had good cytocompatibility and significantly improved cell viability at the early stage of cell proliferation. Figure 5b presents the morphology of MC3T3-E1 cells adhered to an MAO-HA coating and a PDA/MAO-HA coating. It can be clearly observed that cells developed well on the PDA/MAO-HA coatings after 24 h, spreading over the surface, as expected, with prominent lamellipodia extensions. Filopodia can also be found, indicating the good connection between the coating surface and cells. In contrast to the cells cultured on the PDA/MAO-HA coating, cells on the MAO HA coatings grew poorly, and some cells did not exhibit the elongated morphology.

### 2.4. Sol-Gel

Sol-gel is a process through which a network is formed from solution by a progressive change in liquid precursor(s) into a sol, to a gel, and, in most cases, finally to a dry network [48]. The sol-gel method is a kind of surface coating protection technology with simple operation and at a low processing temperature. Sol-gel coating has the advantages of excellent adhesion and good uniformity, which can obviously improve the corrosion resistance and high-temperature oxidation resistance of the substrate in some environments. The sol-gel technique was used to prepare an HA coating on MAO-coated Mg alloys. The thickness of the sol-gel/MAO coating was about 8 μm, and the density of the sol-gel/MAO coating was higher than that of the MAO coating, with a smoother surface than that of the MAO coating. The sol-gel/MAO coating, as a barrier layer, effectively blocked the formation of micropores in the MAO coating. Compared with the MAO coating, it had a lower corrosion current density and higher electrochemical impedance, meaning that it had better corrosion resistance. SEM images of the samples after 6 h of immersion are shown in Figure 6. The degree of corrosion at the interface between the sol-gel/MAO coating and scratch is lighter than that between the MAO coating and scratch, indicating that the sol-gel/MAO coating has better corrosion resistance, which is consistent with the local electrochemical test results [49]. However, the porous structure formed by the Ca-P coating in an aqueous solution could not achieve the ideal corrosion resistance for a long time. Guo et al. [50] prepared a titanium dioxide (TiO_2_) coating on a Mg alloy, which was further coated with HA by the sol-gel method. Electrochemical experiments showed that compared with the HA coating, the HA/TiO_2_ coating could improve the corrosion resistance more effectively. The cell viability of the Mg alloy coated with HA/TiO_2_ was studied by an extraction method. It was found that the cell viability of the Mg alloy coated with HA/TiO_2_ was up to 86.99%. Most of the cells on the surface of the HA/TiO_2_ coating could adhere and diffuse, and the cells were connected with each other, which was conducive to the cell diffusion on the surface of the Mg alloy. Since initial bacterial adhesion and accumulation often leads to biofilm formation, and then implantation failure, it is crucial to prevent the bacterial infection early after implantation. In vitro antibacterial experiments were carried out on the Mg alloy. The antibacterial rates of the composite coated Mg alloy on *Escherichia coli* and *Staphylococcus aureus* reached 99.5 ± 0.3% and 99.8 ± 0.2% after 24 h, respectively, which were much higher than those of the uncoated Mg alloy.

### 2.5. Hydrothermal Synthesis

Hydrothermal synthesis is the synthesis of substances in an aqueous solution, by a chemical reaction, in a temperature range of 100–1000 °C and a pressure range of 1 MPa–1 GPa. An HA coating prepared by a hydrothermal method has high crystallinity and a low dissolution rate.

Wen et al. [51] prepared an HA coating by hydrothermal bonding on the surface of an AZ31B Mg alloy. It was found that when the concentration of calcium phosphate solution was 0.1 mol/L, and the Ca/P ratio was 1.67, the coating showed good morphology, structure, and corrosion resistance. When the Ca/P ratio decreased down to 1.58, the corrosion potential of the prepared coating increased from −1.51 V to −1.18 V, and its impedance reached 1.0 × 10^5^ Ω·cm^2^, which effectively delayed the early corrosion of the Mg alloy substrate. Li et al. [52] pre-deformed a ZEK100 Mg alloy by a high-pressure torsion (HPT) process, and then prepared HA coatings with different contents of Mg(OH)_2_ nano-powders on the surface, by a hydrothermal synthesis method. The adhesive tape test results showed that the HA coating containing 0.3 mg/mL Mg(OH)_2_ had an interfacial bonding strength of 4B (coating peeling area was less than 5%), while that of the untreated Mg alloy was 2B (coating peeling area was 15–35%). There was a large number of fine grains, twins, and grain boundaries in the microstructure of the HPT-treated Mg alloy, which provided more nucleation sites for the HA coating than the Mg alloy. Meanwhile, 0.3 mg/mL of Mg(OH)_2_ nano-powder could promote the deposition of HA and make the coating more compact, and the interface bonding strength of the HA coating was greatly increased. Zhang et al. [53] successfully prepared an HA coating on the surface of a fluorinated AZ31 Mg alloy, by hydrothermal synthesis. The thickness of the HA/MgF_2_ composite coating was about 10 μm, and the corrosion resistance was good. Furthermore, it was found that the bonding strength between the HA coating and Mg alloy matrix was improved by the MgF_2_ interlayer. The bonding between the dense MgF_2_ interlayer and the HA coating isolated the contact between the SBF and the Mg alloy matrix, thus delaying the degradation of the Mg alloy. CCK8 assay and live/dead staining were used to examine the proliferation of MG63 cells on the alloy surface. The HA/MgF_2_-coated nanocrystal structure promoted the early adhesion of cells. After 7 days, the HA/MgF_2_-coated sample was fused with MG63 cells, and the biocompatibility was good [54]. Some studies have shown that the presence of strontium (Sr) has a positive effect on promoting bone formation and regeneration, inhibiting bone resorption and preventing osteoporosis [55,56,57]. Yang et al. [58] prepared a strontium-substituted HA coating (Sr-HAC) on an AZ91D alloy, by a hydrothermal synthesis method. The experimental results showed that Sr-HAC reduced the concentration of released Mg^2+^ ions, and effectively improved the viability of MG63 cells. An in vitro cell culture experiment showed that Sr-HAC, with a nanostructured lamellar surface, significantly increased the viability of osteoblasts and improved the biocompatibility of the Mg alloy. However, it is difficult to deposit an HA coating on a Mg alloy, due to the lack of adsorption sites in hydrothermal synthesis [59]. Zhou et al. [60] induced the formation of an HA coating on the surface of a AZ31 Mg alloy, through an interlayer of PDA. As shown in Figure 7, the dopamine-induced HA coating is thicker and denser than the pure HA coating, and the synergistic effect of the two significantly reduced the corrosion rate. Moreover, cell proliferation was studied by a co-culture of alloy extract and cells, and it was found that the survival rate of cells coated with dopamine HA reached 120% after 5 days. A large number of polygonal cells could be observed on the coated samples, and filopodia existed between the cells and the matrix.

### 2.6. Other Methods

In addition to the above methods, other methods to prepare HA coating on Mg alloys include laser deposition, microwave aqueous-phase synthesis, ultrasonic aqueous-phase synthesis, chemical conversion method, etc. Rau et al. [61] used pulsed laser deposition technology to deposit an HA coating on a Mg-Ga alloy. The results showed that higher temperatures caused the decomposition of HA, and the HA coating deposited at 200 °C and 300 °C showed good corrosion resistance in the simulated body fluids. Shen et al. [62] synthesized a two-layer HA coating on the surface of a Mg alloy, by rapid microwave water-phase synthesis method. The coating consisted of a cotton-like top layer of HA and a strip-like bottom layer of HA. The immersion test indicated that the strip-like HA layer with a thickness of 7.7 μm and low solubility endowed the Mg alloy a protection effect at the initial stage, and the nano-scale cotton-like HA layer exhibited a high HA mineralization capacity. The mineralized layer reached a thickness of 12.7 μm after 24 days of immersion, and eventually integrated with the original HA bilayer coating, providing favorable long-term protection for the Mg alloy. In addition, the potentiodynamic polarization test demonstrated that the corrosion current density of an HA-coated Mg alloy prepared with microwave heating for 10 min was 100 times lower than that of the naked counterpart. Sun et al. [63] prepared an HA coating on a Mg alloy using a new ultrasonic aqueous-phase synthesis method. The specific method was to synthesize a dense crack-free HA coating in an aqueous solution containing Ca^2+^ and PO_4_^3−^ ions, by ultrasonic cavitation for 1 h. The interfacial bonding strength of the coating was 18.1–2.2 MPa. The coating was made of bamboo leaf HA in an irregular arrangement. Compared with the uncoated Mg alloy, the coated alloy exhibited excellent corrosion resistance and rapidly induced apatite formation after soaking in SBF for only 3 days. The HA coating could protect the Mg alloy substrate for a long time after the 90-day immersion test. Hu et al. [64] fabricated a Ag/HA composite coating on an extruded Mg-2Zn-1Mn-0.5Ca alloy, by the chemical transformation method. The corrosion resistance was significantly improved and the adhesion test also showed that the coating had a high bonding strength (55 MPa). The antibacterial activity of the Ag/HA-coated alloy was uniform, and an inhibition zone, about 14 mm in diameter, was found around the Mg alloy, indicating that the Ag/HA composite coating had a significant antibacterial effect on *S. aureus*. You et al. [65] prepared a dense HA coating on a JDBM alloy, by two-step chemical deposition and subsequent hydrothermal synthesis. The bonding strength between the coating and the substrate was about 2 MPa. The coated alloy, after 1 h of hydrothermal treatment, showed significant cell growth. The cells on the surface of the alloy, after 3 and 12 h of hydrothermal treatment, showed a polygonal shape, filopodia, and good cell–cell adhesion. The alkalinity phosphatase assay (ALP) showed that the ALP values of 1, 3, and 12 h hydrothermal treatment were increased by 50.5%, 17.6%, and 29.7%, respectively, compared with the control group, indicating that the coated alloy promoted the osteogenic differentiation of MC3T3-E1 cells and had better biocompatibility.

## 3. Summary

The current developments have shown that biodegradable Mg alloys are indeed a promising class of biomedical materials. However, no single surface protection technology has been developed to protect Mg alloys from rapid corrosion, while having good biocompatibility and bioactivity. In summary, it can be observed that HA coating has a beneficial effect on reducing the corrosion rate of magnesium alloys, which provides the possibility of adjustable degradation. Meanwhile, HA coating also has good biocompatibility, which can be coated on the surface of the cardiovascular stent, bone internal fixation device, and other implants, and applied in vivo in the future. However, the above studies show the high brittleness of the HA material itself, and that the bonding strength between the coating and matrix is low, which also brings challenges to the performance of implants. HA composite coating technologies should become the main development direction in the future, if the bonding properties of coatings are improved. Combining two or more coating technologies, such as sol-gel and micro-arc oxidation, chemical transformation, hydrothermal synthesis, and many more, can enhance the adhesion strength and bioactivity of alloys. In addition, more basic research is needed to deeply reveal the metabolic mechanism of HA-coated Mg alloys and the osteogenic effect in vivo, to provide a scientific basis for the clinical application of degradable Mg alloys.

## Figures and Tables

**Figure 1 materials-14-05550-f001:**
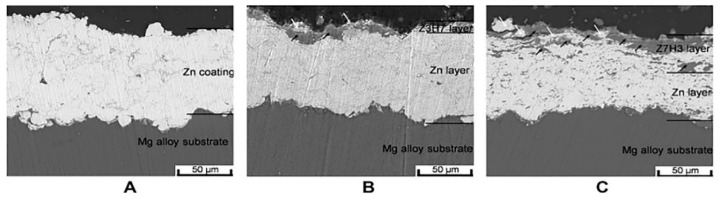
Cross-sectional microstructures of Zn and HA/Zn composite coatings, (**A**) Zn; (**B**) Z3H7; (**C**) Z7H3 (reprinted from Ref. [25] with permission).

**Figure 2 materials-14-05550-f002:**
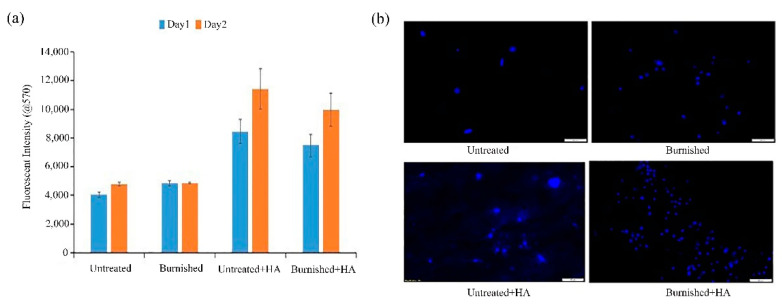
Cell viability (**a**), fluorescence images of cell density of untreated and treated AZ31B alloy samples (**b**) (reprinted from Ref. [28] with permission).

**Figure 3 materials-14-05550-f003:**
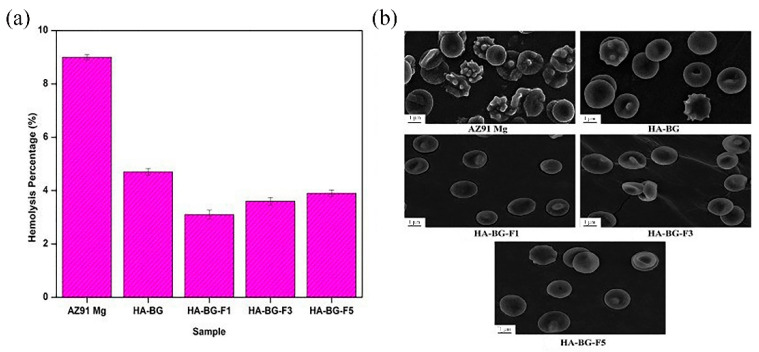
Hemolysis ratio (**a**) and RBCs morphology after exposure (**b**) (reprinted from Ref. [38] with permission).

**Figure 4 materials-14-05550-f004:**
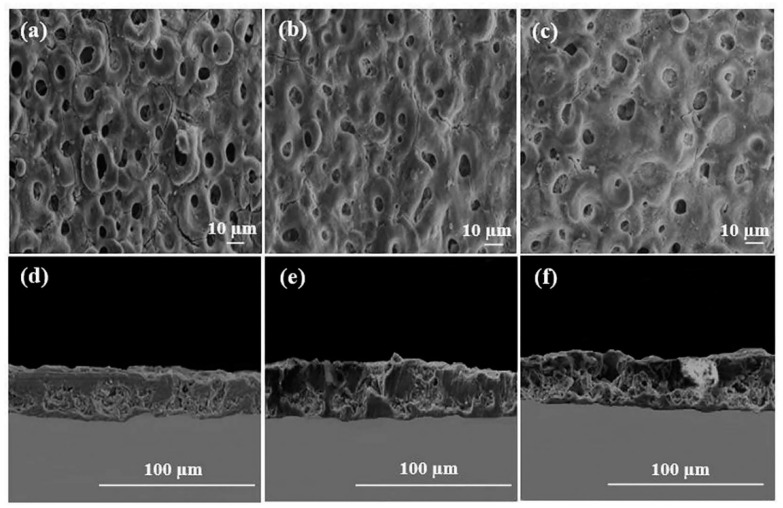
SEM images of cross-sections (**a**–**c**) and surfaces (**d**–**f**) on HA coatings with various concentrations (reprinted from Ref. [42] with permission).

**Figure 5 materials-14-05550-f005:**
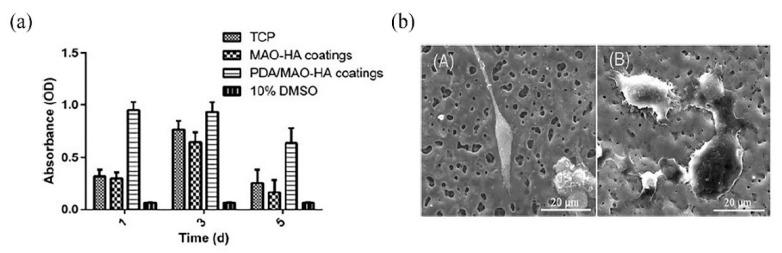
Optical densities (**a**), SEM images (**b**) of MC3T3 E1 cells after culturing for 1 day on the following different coatings: (**A**) MAO HA coating and (**B**) PDA/MAO HA coating (**b**) (reprinted from Ref. [47] with permission).

**Figure 6 materials-14-05550-f006:**
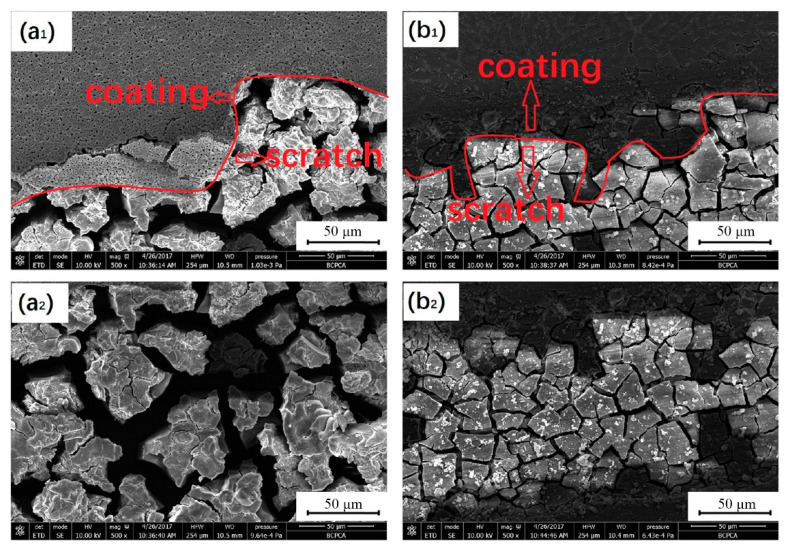
SEM images of the sample after 6 h immersion MAO coating and scratch (**a1**), scratch of MAO (**a2**), sol-gel/MAO coating and scratch (**b1**), scratch of sol-gel/MAO (**b2**) (reprinted from Ref. [49] with permission).

**Figure 7 materials-14-05550-f007:**
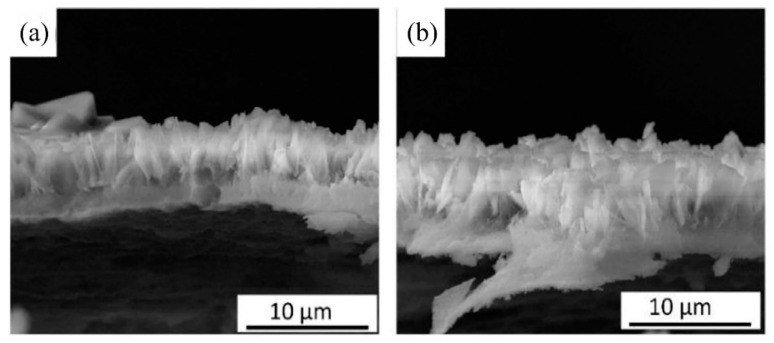
Cross-sectional images of HA coating (**a**) and PDA-HA coating (**b**) (reprinted from Ref. [60] with permission).

**Table 1 materials-14-05550-t001:** Hardness and adhesion strength of different coatings (reprinted from Ref. [32] with permission).

Samples	Microhardness (HV)	Adhesion Strength (Mpa)
Uncoated Mg	37.1 ± 1.5	–
AN-Mg	58.5 ± 1.5	18.2 ± 0.4
DCPD-Mg	66.4 ± 2.2	21.6 ± 0.3
HA-Mg	88.8 ± 2.7	23.3 ± 0.5
Uncoated WE43	76.7 ± 2.5	–
AN-WE43	106.3 ± 3.5	19.6 ± 0.3
DCPD-WE43	120.0 ± 3.3	23.2 ± 0.2
HA-WE43	164.1 ± 4.8	28.5 ± 0.4

## Data Availability

Data sharing is not applicable to this article.

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
