# Peer review of "Recent Advances on Development of Hydroxyapatite Coating on Biodegradable Magnesium Alloys: A Review"

_materials, 2021, doi:10.3390/ma14195550_

Round 1
Reviewer 1 Report
The manuscript materials-1382857 entitled "Recent advances on development of hydroxyapatite coating on biodegradable magnesium alloys: a review" presents the current information on biodegradable magnesium alloys with hydroxyapatite coating.
The introduction fully explains what the article is about and how important hydroxyapatite coatings on magnesium alloy are in biomaterials.
The review of the proposed coatings is structured according to synthesis methods such as: spraying, thermal spraying, cold spraying, electrochemical process, sol-gel and hydrothermal synthesis by others methods.
Authors described each of those synthesis methods and subsequently described the properties of the hydroxyapatite coating thus obtained.
The topic raised is very important. Magnesium (Mg) alloys have good biocompatibility, bioactivity and biodegradability. The problem is the degradation rate of Mg alloy is fast. During this process a highly alkaline environment is generated, leading to the bone lysis. Hydroxyapatite coating is to protect the
Magnesium alloy implant against the corrosion.
Unfortunately, as the authors write in conclusion, the currently used coatings do not provide adequate protection and further research is needed.
This is an interesting work and well-presented. I recommend to accept this manuscript for publication in Materials.
Specific comments:
- I suggest it to clearly inform which Figures have been reprinted e.g. (reprinted from ref. [ ] with permission).
Reviewer 2 Report
The article entitled “Recent advances on development of hydroxyapatite coating on biodegradable magnesium alloys: a review” is an interesting review paper on the development of hydroxyapatite on biodegradable magnesium alloys. The authors have developed a comprehensive report of previous experiences. Only minor English corrections are required. The abstract, section 2 and the conclusion sections require major revision by the authors because they do not include coherent information.
Abstract – The abstract does not provide a meaningful insight into the article. It needs to identify 1) the scope of the article more clearly, 2) critical properties for these materials (adhesion, biocompatibility and bioactivity, mentioned in the conclusion) 3) the current gaps in research/literature and further research directions
Section 2
Further clarification on bioactivity aspects would benefit the article (this could be done with a new section being introduced). This is left to authors’ discretion.
Adhesion and Biocompatibility are more often mentioned in the article, however, there is no clear insight on how this can be quantified or on why this is critical for these materials. Again, new sections could be useful.
Line 341-345 – Please provide a better definition of sol-gel https://goldbook.iupac.org/terms/view/ST07151
Conclusion – Conclusion does not provide a clear hierarchy of priorities of what is more relevant for each aspect of research.
Reviewer 3 Report
The paper is well organized and presented. There are more paper in the field which are not included in this study showing promising results in the field. I think the conclusions should be revised to provide a more accurate pictureof the whole study. Also some.experimental examples for thermal spray coatings are welcome for a better comprehesion of the coating techniques.
Round 2
Reviewer 2 Report
English spelling checks needed only.